# Antimicrobial Potential of Betulinic Acid and Investigation of the Mechanism of Action against Nuclear and Metabolic Enzymes with Molecular Modeling

**DOI:** 10.3390/pathogens12030449

**Published:** 2023-03-13

**Authors:** Gabriela Cristina Soares Rodrigues, Mayara dos Santos Maia, Thalisson Amorim de Souza, Edeltrudes de Oliveira Lima, Luiz Eduardo Carneiro Gomes dos Santos, Shellygton Lima Silva, Marcelo Sobral da Silva, José Maria Barbosa Filho, Valnês da Silva Rodrigues Junior, Luciana Scotti, Marcus Tullius Scotti

**Affiliations:** 1Laboratory of Cheminformatics, Program of Natural and Synthetic Bioactive Products (PgPNSB), Health Sciences Center, Federal University of Paraíba, João Pessoa 58051-900, PB, Brazil; 2Multi-User Characterization and Analysis Laboratory, Program of Natural and Synthetic Bioactive Products (PgPNSB), Health Sciences Center, Federal University of Paraíba, João Pessoa 58051-900, PB, Brazil; 3Program of Natural and Synthetic Bioactive Products (PgPNSB), Health Sciences Center, Federal University of Paraíba, João Pessoa 58051-900, PB, Brazil; 4National Institute of Science and Technology in Tuberculosis (INCT-TB), Porto Alegre 90619-900, RS, Brazil; 5Program of Biotechnology, Center for Biotechnology, Federal University of Paraíba, Campus I, Cidade Universitária, João Pessoa 58051-900, PB, Brazil

**Keywords:** betulinic acid, antimicrobial activity, triterpenoids, *M. tuberculosis*, *E. coli*

## Abstract

Natural products have important pharmacological activities. This study sought to investigate the activity of the compound betulinic acid (BA) against different strains of bacteria and fungi. The minimum inhibitory concentration (MIC) was determined and then the minimum bactericidal concentration (MBC) and minimum fungicidal concentration (MFC). After performing the in vitro tests, molecular modeling studies were carried out to investigate the mechanism of action of BA against the selected microorganisms. The results showed that BA inhibited the growth of microbial species. Among the 12 species (*Staphylococcus aureus*, *S. epidermidis*, *Pseudomonas aeruginosa*, *Escherichia coli*, *Mycobacterium tuberculosis*, *Candida albicans*, *C. tropicalis*, *C. glabrata*, *Aspergillus flavus*, *Penicillium citrinum*, *Trichophyton rubrum*, and *Microsporum canis)* investigated, 9 (75%) inhibited growth at a concentration of 561 µM and 1 at a concentration of 100 µM. In general, the MBC and MFC of the products were between 561 and 1122 μM. In silico studies showed that BA presented a mechanism of action against DNA gyrase and beta-lactamase targets for most of the bacteria investigated, while for fungi the mechanism of action was against sterol 14α-demethylase (CYP51) targets and dihydrofolate reductase (DHFR). We suggest that BA has antimicrobial activity against several species.

## 1. Introduction

The spread of infections resistant to available antimicrobial drugs is a major threat to human health [1]. According to the WHO, in less than 30 years, microbial resistance could become the leading cause of death. To avoid microbial resistance, the identification of new targets and innovative drugs with new modes of action that are able to evade resistance mechanisms may represent a valid solution to counteract the continuous emergence and spread of resistant infections [2] Mapping the spread of resistance, the development of rapid diagnostics to ensure early selection of appropriate therapies, incentives for discovery of new agents, and studies of therapeutic pathways are alternatives to minimize cases of resistance [3].

Triterpenoids are one of the largest subclasses of terpenoids, with over 14,000 known structures [4]. These structures are cyclized from squalene precursors oxidized by oxidosqualene cyclases, with wide structural diversity ^4^. Most triterpenes, predominantly pentacyclics, are widely distributed in plants—in seeds, stem bark, roots, leaves, or the wax-like coating of various fruits and herbs such as thyme, mistletoe, or lavender [5]. Betulinic acid (BA) (Figure 1) is a lupane-type pentacyclic triterpenoid that has a variety of medicinal properties [6]. BA has gained prominence and considerable research interest because of its potent physiological and pharmacological activities [7]. It has been reported to have a variety of effects, including antitumor [8], anti-inflammatory [9], anti-HIV [10], antidiabetic [11], antibacterial [12], antiviral [13], and antimalarial [14] activities, among others. BA can be isolated from several plants, such as *Quisqualis fructus*, *Coussarea paniculata*, *Caesalpinia paraguariensis*, *Vitex negundo*, *Ilex macropoda*, *Anemone raddeana, Doliocarpus schottianus*, *Tovomita krukovii*, *Chaenomeles lagenaria*, *Berlinia grandiflora Vietnamese*, *Orthosiphon stamineus*, and *Eucalyptus*, among others [15].

Phytochemicals, due to their various pharmacological properties, can act against bacterial and fungal resistance, as reported in several studies [16,17]. These agents can act alone or in combination with antibiotics to increase antibacterial activity against a wide range of bacteria [18]. As for antifungal agents, although new mechanisms of action are at various stages of clinical evolution, their number is relatively small compared to other diseases [19]. Consideration of the common structural features distinguishing fungi should facilitate the development of antifungal drugs with broad-spectrum activity, and the outcome of systemic fungal infections strongly depends on how quickly treatment is initiated [20].

Most compounds used in infectious disease treatments are either natural or semi-synthetically modified to improve efficacy by targeting the engineering of natural products into potent antibiotics [21]. Therefore, it is necessary to screen natural products that may be new therapeutic alternatives for the treatment of infectious diseases [17]. Among all available options, plant-derived compounds have shown the most potential applications in fighting bacterial and antifungal infections [18]. 

According to Matamoros-Recio et al. [22], it is very difficult to study microorganism structures at the molecular level by experimental techniques, despite the significant development of fast and efficient experimental protocols. Due to the various cases of resistance, it is important to study new routes and new targets and develop new compounds capable of overcoming the factors that contribute to antimicrobial resistance. Therefore, the authors emphasize that computational methods allow the resolution of atomic structures and reveal molecular aspects (interatomic and intermolecular interactions) not accessible by any experimental microscopy. This information significantly contributes to reducing costs in the production of antimicrobial drugs that have high potential. Computational methods contribute to the discovery of more selective, potent, and efficient drugs. Thus, our study sought to investigate the potential of BA against various bacterial and fungal strains through in vitro activity, as well as molecular-level analysis of the likely mechanisms of action through computational studies.

## 2. Materials and Methods

### 2.1. Extraction of Betulinic Acid

Betulinic acid was extracted from the bark of *Zizyphus joazeiro* Mart. (Rhamnaceae) using a method described earlier [23,24] by Barbosa et al. (1985), using column chromatography for the extraction and treatment of acid hydrolysis. The isolation was carried out by comparing physical properties obtained through various spectrometric methods (mp, if, uv, ms, ^1^H nmr). 

### 2.2. Quantitative Structure–Activity Relationship Modeling (QSAR)

Among the species of bacteria and fungi investigated in this study, we were able to build a predictive model of biological activity against the species *Escherichia coli* and a model already reported in the literature against *Mycobacterium tuberculosis* [25]. For the other species, it was not possible to build predictive models due to lack of data or low amount of data and/or because we were unable to validate some models.

Knime 3.6.2 software (Knime 3.6.2, Copyright Miner, by Konstanz Information, Zurich, Switzerland, www.knime.org, accessed on 14 August 2022) was used to perform the QSAR modeling. The compounds were saved in SDF format and imported into the Dragon 7.0 software (Kode Chemoinformatics srl, Pisa, Italy) [26] for generating descriptors. The random forest (RF) algorithm was used to build the *E. coli* model, according to the procedures described in [27]. External cross-validation was performed to estimate the predictive power of the developed models. In addition, the performance of external models was evaluated by ROC analysis and Matthews’s coefficient (MCC) [28] to evaluate the model globally.

### 2.3. Antimicrobial Activity Tests

#### 2.3.1. Preparation of Compounds

BA and antimicrobials (controls) obtained from Sigma-Aldrich/Merck KGaA (Darmstadt, Germany) were subjected to biological assays to evaluate their antimicrobial activity against strains of bacteria, yeast, and filamentous fungi. The compounds were weighed and properly solubilized in 150 µL (5%) of dimethyl sulfoxide ((DMSO) Labsynth—Labsynth Products for Laboratories Ltd., Sao Paulo, Brazil), and 60 µL (2%) of Tween 80 (InLAB—scientific technical materials Ltd., Sao Paulo, Brazil) was added. The final volume was made up with sterile distilled water q.s.p. 5 mL, obtaining an initial concentration of 1024 µg/mL, and serially diluted to 4 µg/mL [29,30].

For the antimicrobial activity against the bacterium *Mycobacterium tuberculosis* H37Ra, BA was first solubilized in DMSO at a concentration of 8 mM and then diluted in Middlebrook 7H9 + 10% OADC broth to reach a concentration of 400 µM.

#### 2.3.2. Microorganisms

Most microorganisms belonged to the Micoteca of the Mycology Laboratory, Federal University of Paraíba (UFPB), and the following strains were used for the biological assays:

Bacteria: *Staphylococcus aureus* ATCC-13150, *S. epidermidis* ATCC-12228, *Pseudomonas aeruginosa* ATCC-25853, *Escherichia coli* ATCC-18739. Yeast fungi: *Candida albicans* ATCC-90028, *C. albicans* LM-34, *C. tropicalis* ATCC-13803, *C. tropicalis* LM-77, *C. glabrata* ATCC-90030. Filamentous fungi: *Aspergillus flavus* ATCC-13013, *Penicillium citrinum* ATCC-40011, *Trichophyton rubrum* LM-34, *Microsporum canis* LM-12.

The *M. tuberculosis* H37Ra strain belonged to the UFPB Biotechnology Center.

#### 2.3.3. Minimum Inhibitory Concentration (MIC)

The minimum inhibitory concentration was determined using the protocols of the Clinical and Laboratory Standards Institute [31]. The cultures of bacterial strains in brain heart infusion (BHI) medium and fungi in agar Sabouraud dextrose (ASD) medium were suspended in sterile 0.9% saline solution and adjusted according to the 0.5 tube of the McFarland standard scale to obtain 106 CFU/mL [32,33]. The microdilution was performed in a plate containing 96 wells, distributing 100 μL BHI broth/bacteria and 100 μL of RPMI 1640 broth/fungi in the wells, followed by serial dilution [34]. The prepared plates were aseptically closed and incubated at 35 ± 2 °C for 24–48 h for bacterial and yeast assays and at RT (28–30 °C)/5–7 days for filamentous fungi.

The MIC for BA against *M. tuberculosis* H37Ra was as previously reported [35,36]. Twofold serial dilutions were performed in 96-well U-bottom polystyrene microplates in concentrations ranging from 200 to 3 µM. Mycobacterial suspensions were grown at 37 °C and diluted in 7H9 + 10% OADC medium to a theoretical optical density (OD600nm) of 0.006, and 100 µL was added to each well [36]. After 7 days of incubation at 37 °C, 30 µL of a resazurin solution (0.02%) was added to the plates, and the results were evaluated after 48 h [36]. MICs were determined using the resazurin reduction microplate assay (REMA) as an indicator of growth and were considered to be the lowest concentration of drug that prevented a color change from blue (resazurin) to pink (resorufin). The values reported here were observed in three independent experiments. Rifampicin and moxifloxacin were used as positive control drugs.

#### 2.3.4. Minimum Bactericidal Concentration (MBC) and Minimum Fungicidal Concentration (MFC)

After reading the MIC, the determination of MBC and MFC was performed, removing aliquots from the microdilution plates in the corresponding wells and inoculating them into new plates containing only culture broth. All controls were performed in parallel. MBC is defined as the lowest concentration capable of causing complete inhibition of bacterial growth after 24 h at 35 °C [37]. MFC was considered to be the lowest concentration of the test product capable of inhibiting the growth of microorganisms after 24–48 h at 35 ± 2 °C/yeasts and RT (28–30 °C)/5–7 days for filamentous fungi [38,39].

### 2.4. Molecular Modeling

#### 2.4.1. Alignment of Protein Sequences

Sequence alignment was used to assess the degree of identity between the sequences of the proteins selected in this study among the species investigated. This analysis is important for the investigation of conserved regions, identification of the active site, and construction of three-dimensional proteins that have not yet been structurally resolved. Three types of proteins were chosen for bacteria (DNA gyrase, beta-lactamase, and penicillin-binding protein (PBP)) and three types of enzymes for fungi (sterol 14α-demethylase (CYP51), secreted aspartic protease 2 (SAP- 2), and dihydrofolate reductase (DHFR)). Initially, the protein sequences of the selected species were obtained from the GenBank database (National Center for Biotechnology Information, Bethesda, MD, USA, https://www.ncbi.nlm.nih.gov/, accessed on 14 August 2022) [40]. They were then aligned using the Clustal Omega web tool (https://www.ebi.ac.uk/Tools/msa/clustalo/, accessed on 14 August 2022) [41] which aligns all user-entered protein sequences. Access codes for protein sequences are described in Appendix A. Sequences missing from the table are due to the lack of information on genomic and proteomic sequencing in the databases.

#### 2.4.2. Modeling by Homology

Homology modeling was used to build three-dimensional protein structures of proteins that are not yet available in a database. The target sequences were obtained as amino acid sequences in FASTA format from the GenBank website and then imported into the SWISS-MODEL website (https://swissmodel.expasy.org/, accessed on 14 August 2022) to build the models [42]. For each identified mold, the quality was predicted from alignment features such as ProMod3, QMEAN, and GMQE. The identification of the templates and the degree of identity of the target and template sequences are described in Appendix A. For the enzymes of *Microsporum canis*, *Penicillium citrinum*, and some enzymes of *Trichophyton rubrum*, it was not possible to build homology models due to lack of genomic data.

The stereochemical quality of the models was evaluated by the PSVS (protein structure validation software suite) web server (http://psvs-1_5-dev.nesg.org/, accessed on 14 August 2022), using PROCHECK. PROCHECK generates a Ramachandran plot [43], which determines the allowed and disallowed regions of the main chain of amino acids. Structural quality was evaluated using the Verify 3D software (saves @ 2020—DOE-MBI Services, http://services.mbi.ucla.edu/SAVES/, accessed on 14 August 2022) [44] and the compatibility between the protein sequence and its 3D structure, based on the chemical environment.

#### 2.4.3. Molecular Docking

Molecular docking was used to investigate the BA mechanism of action that contributes to the antimicrobial effect in various species of bacteria and fungi through the binding affinity of the compound and the enzymes selected in the study. The 3D structures of the selected enzymes were obtained from the Protein Data Bank (PDB) (https://www.rcsb.org/pdb/home/home.do, accessed on 14 August 2022) [45]. Some enzymes were constructed using the homology modeling method, and others could not be constructed due to lack of sequencing or low identity with the target sequence. The selected proteins and detailed information about them can be seen in Table 1. Initially, all water molecules were removed from the crystalline structure, and the mean square deviation (RMSD) was calculated from the postures, indicating the degree of reliability of the fit. RMSD less than 2.0 Å was considered successful. We used Molegro Virtual Docker v.6.0.1 (Molexus IVS, Odder, Denmark) software [46] with the parameters preset in the same software. The complexed ligand was used to define the active site. Standard drugs with the mechanism of action on selected enzymes were chosen as controls. The MolDock SE (Simplex Evolution) [47] algorithm and MolDock Score (GRID) scoring function were used to calculate docking energy values.

## 3. Results

### 3.1. QSAR Modeling

The predictive model against *M. tuberculosis* was validated as reported by Cavalcanti et al. [25]. The predictive model against *E. coli* was evaluated for its predictive powers. The results showed excellent model performance, as represented by the cross-validation values: accuracy = 0.78; specificity = 0.82; sensitivity = 0.72; PPV = 0.78; NPV = 0.78; ROC = 0.86 (Figure 2); and MCC = 0.55. After validation, BA was subjected to predictive analysis in the built models. The results showed that BA was within the chemical domain and presented biological activity of 63% against *M. tuberculosis*, that is, the potential to inhibit the growth of *M. tuberculosis*. While BA showed 65% biological activity against *E. coli*.

### 3.2. Antimicrobial Activity Tests

The antimicrobial activity results of BA and the controls are recorded in Table 2, Table 3 and Table 4.

BA at a concentration of 561 µM caused the growth inhibition of 9/12 (75%) microbial species including bacteria and fungi, and the respective MIC was then considered. In parallel, the controls also presented MIC at a concentration of 561 µM in the proportion of 50%/bacteria and 75%/fungi. In general, the MBC and MFC of the products were between 561 and 1122 µM, except for *M. tuberculosis*. The MIC value of BA against *M. tuberculosis* was 100 µM. The MIC values for the controls were: MIC = 0.19 µM for moxifloxacin and MIC = 0.15 µM for rifampin. Therefore, the investigated compound had inhibitory activity against *M. tuberculosis* but was not potent when compared to controls.

The mode of action of the test substance was also evaluated through the MBC/MIC ratio, a methodology used by Hafidh [39] to specify the nature of the antimicrobial effect, considered as bactericidal and fungicidal when the MBC/MIC and MFC/MIC ratio is between 1:1 and 2:1. On the other hand, if the ratio is greater than 2:1, the mode of action is more likely to be bacteriostatic and fungistatic. The results showed that BA had a bacteriostatic effect against *E. coli* and no fungistatic effect for the investigated strains.

### 3.3. Alignment of Protein Sequences

The alignment of protein sequences contributed to identifying the conserved regions and common residues of the active site in enzymes from selected species. Initially, DNA gyrase protein sequences were evaluated among all bacterial species investigated in this study. It was observed through the alignment that bacterial DNA gyrases share 45.39 to 88.26% identity with each other (Appendix A). DNA gyrases from *S. aureus* and *S. epidermidis* share the highest degree of identity at 88.26%. C-terminal regions with low identity that were not part of the active site were excluded from the alignment. According to Broeck et al. [48], the amino acids Val70, Ile74, Asp88, and Met120 are part of the active site of *E. coli* DNA gyrase. We can see from Appendix A that 50% of the site’s amino acids are conserved in all aligned sequences. For the beta-lactamase enzyme, the alignment showed identity of 14.66 to 95.38% (Appendix A). Only bacteria belonging to the same genus of Staphylococcus showed a higher degree of identity between the aligned sequences (95.38%), while most showed a low degree of identity, evidencing the need to design selective inhibitors for each type of bacterial strain. C- and N-terminal regions with low identity were excluded. The alignment of PBP protein sequences showed a degree of identity ranging from 19.31% to 82.63%, with the highest degree of identity remaining for the *Staphylococcus genus* (Appendix A). These results suggest that the DNA gyrase enzyme is the most conserved among the analyzed proteins and that it is necessary to develop selective inhibitors based on the knowledge of the target structure. In addition, the alignment allowed the identification of conserved regions of *S. epidermidis* proteins.

Among the enzymes selected for the fungal species investigated in this study, we noticed, through the protein sequence alignment analyses, that the three enzymes present a greater degree of identity between the species. This is due to the fact that most of the analyzed species belong to the same genus, *Candida*. The aligned sequences of the CYP51 enzyme showed a degree of identity ranging from 47.14 to 82.95%. Among species of the same genus, the results showed that the CYP51 enzyme from *C. glabrata* shares 64.30% identity with *C. albicans* and 63.92% with *C. tropicalis*. These values are considered low for species of the same genus, showing that there is a need to develop drugs for the target of each species. Moreover, *C. albicans* shares 82.95% of its amino acids with *C. tropicalis* (Appendix A). The results also showed that most active site amino acids are conserved across species. Among *Candida* species, we observed two active site amino acid substitutions. The residues Phe58 and Ala62 in *C. albicans* and *C. tropicalis* are replaced by Val and Ile in *C. glabrata*, respectively. According to Hargrove et al. [20], *C. glabrata* is more resistant to the drug fluconazole, and small substitutions and/or mutations in the active site can confer drug resistance.

Due to the lack of genomic sequencing data for the SAP-2 enzyme for most of the fungal species in this study, it was possible to align only the sequences of *C. albicans* and *C. tropicalis*. The results showed that both share only 47.21% identity (Appendix A). These data indicate that the SAP-2 enzyme of the two species are sequentially quite different. Therefore, the development of inhibitors for these enzymes may have different effects for both species. According to Cultffiel et al. [49], the A70450 inhibitor binds to SAP2 20 times stronger than pepstatin, and the amino acids Ile82, Tyr84, Gly85, Asp86 and Ser88 make many contacts along the inhibitor.

The alignment results for the DHFR enzyme showed that the sequences share between 27.84 and 81.25% identity. *Trichophyton rubrum* and *A. flavus* are the species with the enzyme sequences with the lowest identity when aligned with the other species. Another interesting result is that the C. glabrata DHFR shares only 47.37% of residues with *C. albicans* and 45.03% of residues with *C. tropicalis* (Appendix A). However, *C. albicans* and *C. tropicalis* share 81.25% of amino acids.

### 3.4. Modeling by Homology

In this study, seven homology models were constructed for the species: *S. epidermidis* (DNA gyrase, beta-lactamase, and PBP); *C. tropicalis* (SAP-2 and DHFR); *A. flavus* (CYP51); and *T. rubrum* (DHFR).

The Ramachandran diagram is one of the most used tools to evaluate the stereochemical quality of three-dimensional protein structures. The diagram represents all possible combinations of dihedral angles Ψ (psi) versus φ (phi) for each amino acid in a protein, with the exception of glycine, which has no side chains. Models that obtain more than 90% of the amino acids present in the allowed and/or favored regions (colored regions of the graph) are considered reliable (Appendix A). White regions represent outliers, which have bad contacts. The results showed that all the models generated had more than 97% of amino acids in the allowed and favored regions (Appendix A). We also used the Verify 3D software to analyze the compatibility of the three-dimensional (3D) structure with its one-dimensional (1D) amino acid sequence. The results showed that all models had a 3D-1D score of more than 80% (Appendix A), which is considered ideal for an optimal structure.

### 3.5. Molecular Docking

BA was subjected to molecular docking in six types of enzymes to investigate antimicrobial activity against bacteria and fungi. The docking results were generated using the MolDock score function. More negative values indicated better predictions. The protein in which the compound obtained binding energy values higher or close to the standard drug was considered promising for antimicrobial activity.

The generated docking results were validated by redocking the crystallographic ligand with all investigated proteins. The root mean square deviations (RMSDs) of the obtained fit poses were calculated in comparison with the crystal structure. RMSD values less than 2 Å indicate an optimal degree of screening reliability. Information about the starting structures and the results of the redocking validation is shown in Appendix A. The redocking analysis showed that all RMSD values were below 2.0 Å, that is, the generated poses positioned the ligand correctly in the active location. Therefore, the values were considered satisfactory for docking validation. Missing RMSD values are due to the lack of complexed ligands.

The docking results can be seen in Table 5 and Table 6. According to the results, of all proteins analyzed for the species of bacteria investigated, BA presented higher binding energy values than the standard drug for the three types of enzymes (Table 5). We observed that the binding affinity of BA was better for the DNA gyrase and beta-lactamase enzymes. The beta-lactamase of all bacteria was able to present higher binding affinity values than the drug used as a control. According to these results, we suggest that BA has DNA gyrase and beta-lactamase as its mechanism of action.

When analyzing the docking results for the fungal species, we observed that BA was able to interact with CYP and DHFR (Table 6). BA showed higher binding energy values than the standard drug, except for the DHFR of *C. albicans*. These results suggest that the mechanism of action of BA in fungi is on CYP51 and DHFR.

We analyzed in detail the interactions formed by BA with the proteins in which this compound obtained superior binding affinity compared to the standard drug. For the DNA gyrase enzyme, three species obtained excellent results. BA showed binding energy values of −109.22 kcal/mol^−1^ in *S. aureus* DNA gyrase, while the standard drug showed −98.76 kcal/mol^−1^. A hydrophobic interaction of BA with the amino acid Arg458 of the DNA gyrase active site was observed. The bacteria *E. coli* and *M. tuberculosis* also showed higher binding affinity values than the drug. 

The docking results showed that BA presented higher binding energy values than the drug used as a control in all beta−lactamases of the investigated bacteria. We report here the bindings and interactions that contributed to the binding affinity of the compound with the active site of the enzymes. This was the only species among the investigated bacteria that obtained a binding energy value considered satisfactory for a probability of effect through this mechanism of action. The result showed that BA had a binding energy value (−107.59 kcal/mol^−1^) close to the energy value of the standard drug (−113.99 kcal/mol^−1^). 

For fungi, the docking results showed that BA has CYP51 and DHFR as its mechanism of action. Among the fungal species investigated in this study that had genomic and structural data for these enzymes for docking, four species showed excellent docking results for each of the analyzed enzymes, but not SAP−2. The CYP51 enzymes from *C. albicans*, *C. tropicalis*, *C. glabrata*, and *A. flavus* were able to interact well with BA, presenting binding energy values higher or close to the energy value of the drug used as a control. We analyzed the interactions of BA with the DHFR enzyme of the fungal species that obtained excellent molecular docking results. 

Interactions between the enzymes and BA compounds and controls can be seen in Figure 2, Figure 3, Figure 4, Figure 5, Figure 6 and Figure 7. Descriptions of these interactions can be seen in Table 7 and Table 8.

## 4. Discussion

Triterpenoids are widely distributed in the plant kingdom and are known to have many beneficial effects such as anti−inflammatory, immunomodulatory, antiproliferative, and antimicrobial activity [50]. 

Studies have reported the use of triterpenoids as anti−staphylococcal compounds but also as resistance−modifying agents when combined with common antibiotics. Among these studies, we highlight the work of Júnior et al. [51], who verified the ability of BA to reduce the activation of the SOS response and its associated phenotypic changes, induced by ciprofloxacin in *S. aureus*. The result showed that there was no antimicrobial activity against *S. aureus*; however, it was able to reduce ciprofloxacin−induced recA expression.

Furthermore, it was observed that BA inhibited the progress of tolerance and mutagenesis induced by this drug. Another study, by Chung [52], showed that *S. aureus* is sensitive to pentacyclic triterpenoids, particularly α−amyrin (AM), BA, and betulinaldehyde (BE), acting synergistically and through different targets of conventional antibiotics. Thus, triterpenoids can be considered therapeutic alternatives against infections caused by *S. aureus*.

A study by Haque et al. [53] sought to investigate betulin and 51 derivatives thereof against five strains, including *Enterobacter aerogenes*, *E. coli*, *Enterococcus faecalis*, *P. aeruginosa*, and *S. aureus*, and the fungal strain *C. albicans*, using broth microdilution assays. Primary antimicrobial screening at a concentration of 50 mM led to the identification of five triterpenoids with antimicrobial properties (0.70% growth inhibition against one or more microbial strains). However, betulinic acid was not able to inhibit the growth of any of the strains when compared to the controls used in the study. Procedures such as the type of strains and control drugs used can contribute to the production of contradictory results.

Our study also evaluated the mechanism of action on important bacterial and fungal targets that are already considered clinically relevant for the proliferation and survival of these microorganisms. According to Germe et al. [54], DNA gyrases are composed of two subunits forming heterotetrameric complexes that establish a transient double−stranded DNA break and facilitate the passage of DNA polymerase across the break. They are considered clinically validated targets for antibiotics and are essential for the transcription and segregation of the genome in bacterial cells. According to Pozzi et al. [55], beta−lactamases represent the main mechanism of resistance to β−lactam antibiotics, conferring resistance to penicillins and cephalosporins. The role of β−lactamases in bacterial resistance is complex and extensive, but the development of inhibitors of this target is important to obtain the antimicrobial effect.

In another study, carried out by Innocente et al. [56], derivatives obtained from ursolic and betulinic acid were tested against various strains of filamentous fungi and yeasts (*C. albicans*, *C. krusei*, *C. glabrata*, *C. tropicalis*, *C. parapsilosis*, and *Epidermophyton floccosum*, among others). The results of the expressed susceptibility assays showed that compound 10 was active against yeast and fungal filaments, while compound 9 showed antifungal activity only against yeasts. The researchers observed that the most active derivatives had the piperazinyl C−28 substituent. Furthermore, based on the MIC and MFC, *C. glabrata* and *C. tropicalis* were more susceptible than other tested species. In addition, based on the MIC and MFC, *C. glabrata* and *C. tropicalis* were more susceptible than other species tested. According to Hargrove et al. [57], *C. albicans* CYP51 prefers its natural substrate lanosterol but is more strongly inhibited by clinical azoles that have a longer side chain arm. The authors further conclude that the interaction with the enzyme is strengthened by hydrogen bonding with His−377, which is conserved in all CYP51 enzymes of the genus *Candida*.

Infections caused by bacteria and fungi have been worrying millions of people around the world due to the large number infected, opportunistic diseases, and the several cases of antimicrobial resistance. Thus, it is necessary to develop new compounds with potential, selective, and effective antimicrobial activity. For this, natural products are important sources of new therapeutic agents due to their structural diversity.

## 5. Conclusions

Our study showed that bacteria share at least 45.39% of their DNA gyrase amino acids. A lower degree of identity was found for the beta−lactamase and PBP enzymes, with at least 14.66 and 19.31% identity between the bacteria, respectively. As for fungi, we observed a minimum degree of identity of 47.14% for the CYP51 enzyme, 47.21% for the SAP−2 enzyme, and 27.84% for the DHFR enzyme. In general, for species of the same genus, the degree of conserved regions exceeds 80%, except for the species *C. glabrata*, which has protein sequences very different from the enzymes of other species of the genus Candida. These data show that despite these enzymes being present in several microorganisms, they differ sequentially and structurally as the degree of identity and similarity between the sequences decreases. This indicates that it is necessary to plan and develop selective drugs for each target species.

Two predictive models were built against *M. tuberculosis* and *E. coli*. The compound presented a prediction of biological activity of 63 and 65% against *M. tuberculosis* and *E. coli* species, respectively. Given these results, it was possible to investigate the in vitro activity, which was confirmed and extended to other species.

BA at a concentration of 561 to 1122 μM produced growth inhibition of 50%/bacteria and 75%/fungi. When evaluating the bactericidal and fungicidal effect, the results showed that BA had a bacteriostatic effect against *E. coli* and no fungistatic effect for the investigated strains. Thus, we suggest that BA has potential antimicrobial activity.

The construction of protein models and molecular docking made it possible to investigate the mechanism of action of BA against the investigated microorganisms. From the results, we suggest that the BA compound has a mechanism of action against DNA gyrase and beta−lactamase targets for most investigated bacteria, and CYP51 and DHFR for most fungi.

## Figures and Tables

**Figure 1 pathogens-12-00449-f001:**
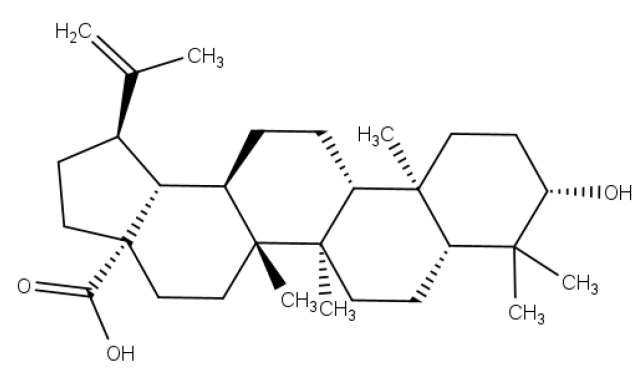
Structure of betulinic acid (BA).

**Figure 2 pathogens-12-00449-f002:**
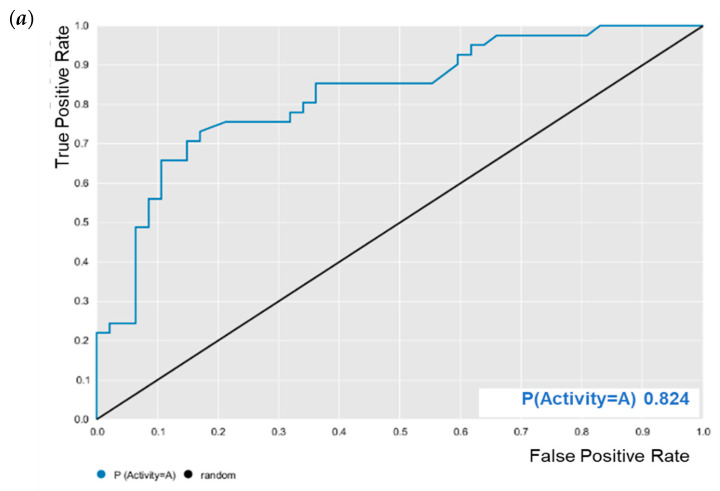
Receiver operating characteristic (ROC) curve generated for *E. coli* random forest (RF) model. (**a**) Test and (**b**) cross-validation.

**Figure 3 pathogens-12-00449-f003:**
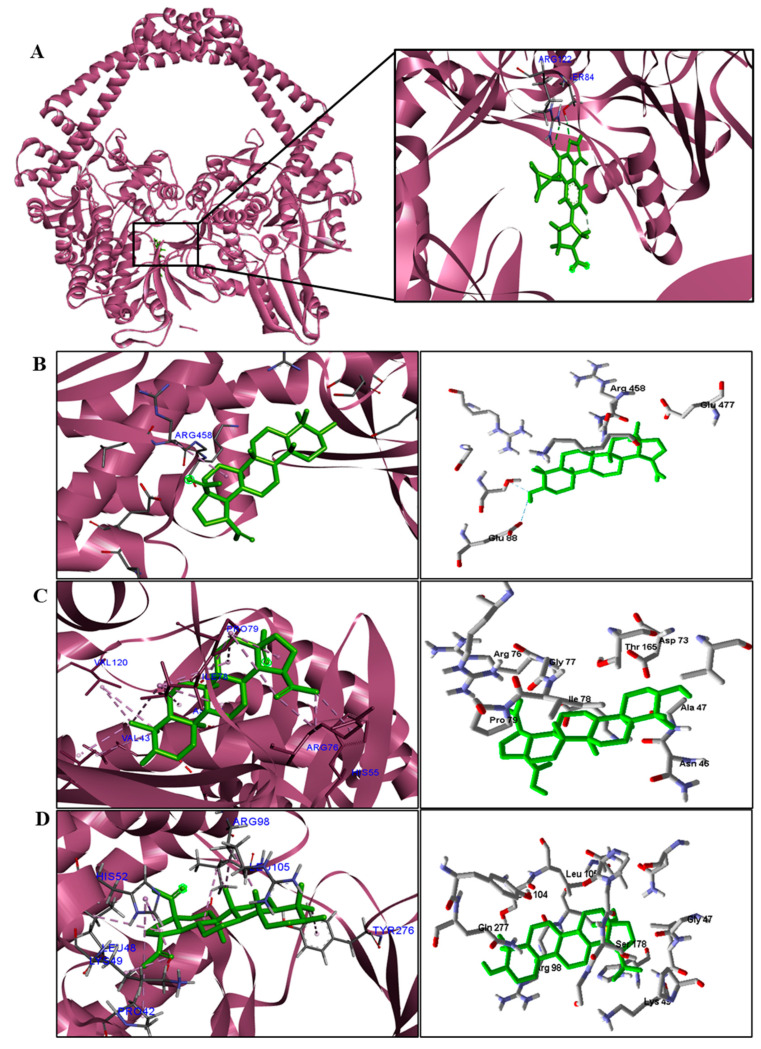
Three−dimensional interactions between BA and DNA gyrase enzyme (**A**) from three bacterial species ((**B**)—*S. aureus*, (**C**)—*E. coli*, and (**D**)—*M. tuberculosis*). Van der Waals interactions are highlighted in light green; hydrophobic interactions in pink; and steric interactions in red.

**Figure 4 pathogens-12-00449-f004:**
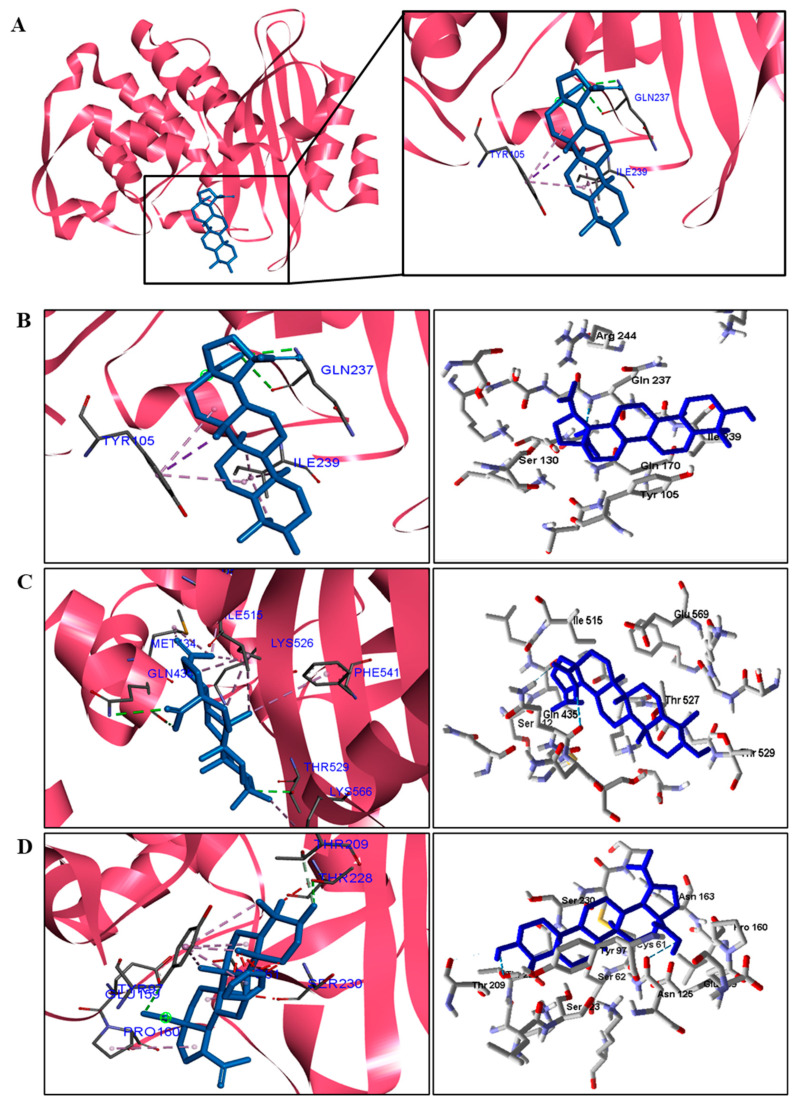
Three−dimensional interactions between BA and the beta−lactamase enzyme (**A**) of three bacterial species ((**B**)—*S. aureus*, (**C**)—*S. epidermidis*, (**D**)—*P. aeruginosa*, (**E**)—*E. coli*, and (**F**)—*M. tuberculosis*). Hydrogen bonds are highlighted in dark green; van der Waals interactions in light green; hydrophobic interactions in pink; and steric interactions in red.

**Figure 5 pathogens-12-00449-f005:**
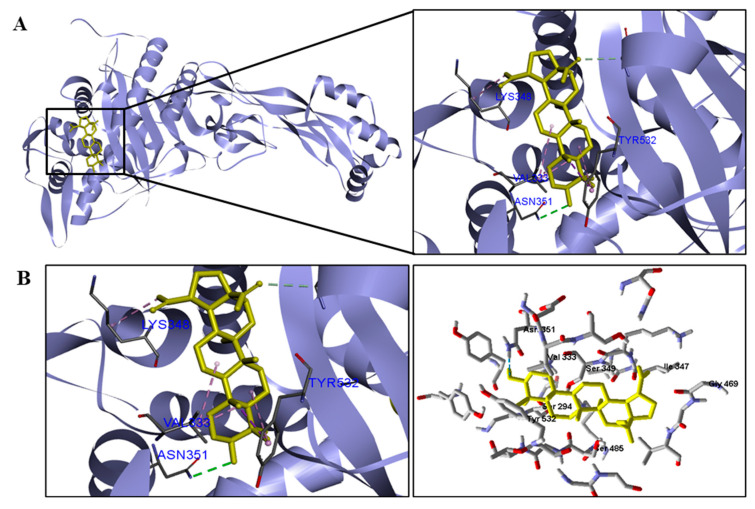
Two− and three−dimensional interactions between BA and PBP enzyme (**A**) and *P. aeruginosa* (**B**). Hydrogen bonds are highlighted in dark green; van der Waals interactions in light green; and hydrophobic interactions in pink and lilac.

**Figure 6 pathogens-12-00449-f006:**
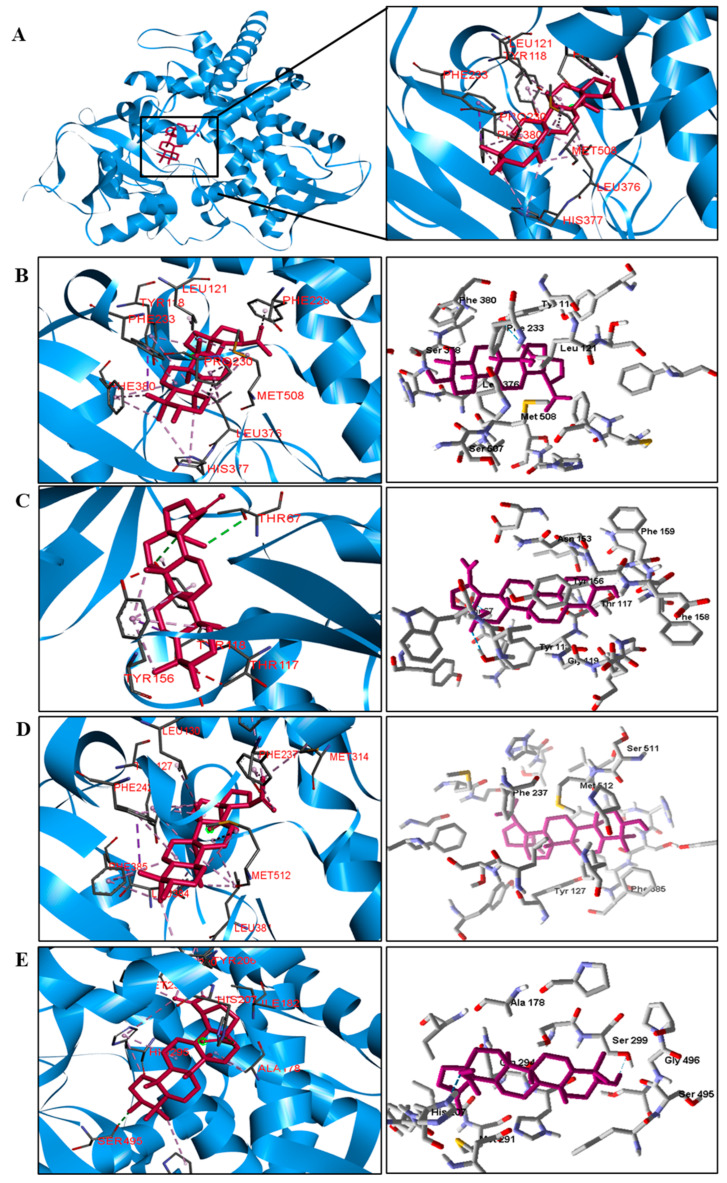
Two− and three−dimensional interactions between BA and CYP51 enzyme (**A**) from four types of fungi (**B**—*C. albicans*, **C**—*C.tropicalis*, **D**—*C. glabrata*, and **E**—*A. flavus*). Hydrogen bonds are highlighted in dark green; van der Waals interactions in light green; and hydrophobic interactions in pink and lilac.

**Figure 7 pathogens-12-00449-f007:**
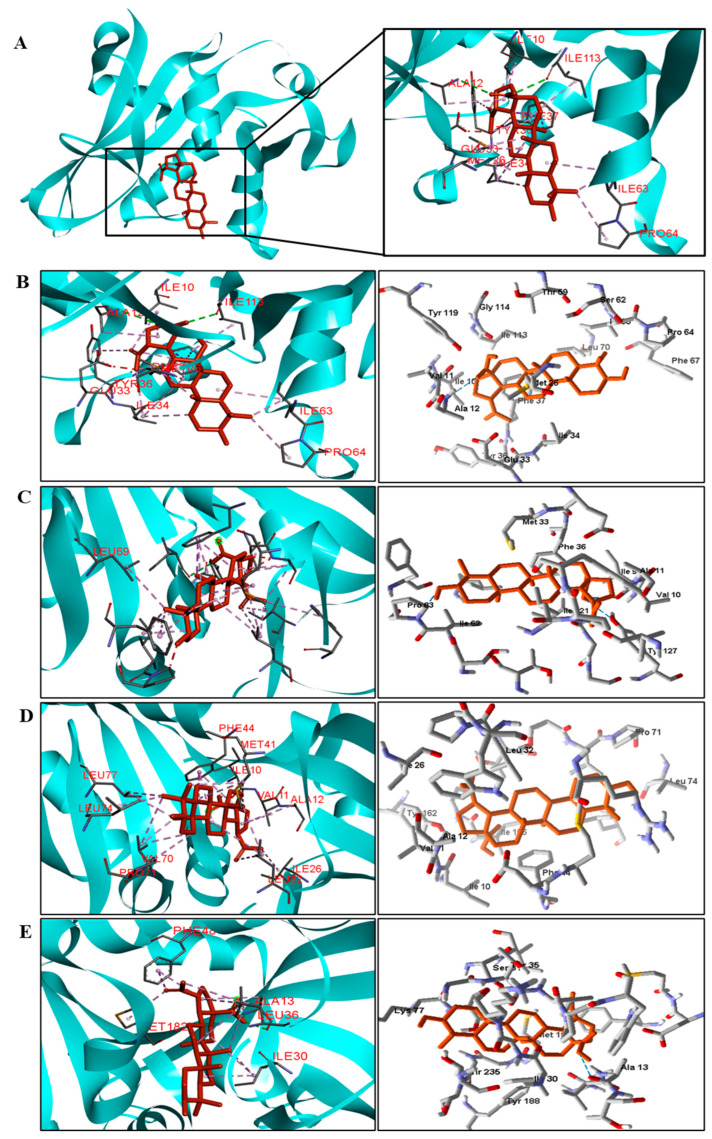
Two− and three−dimensional interactions between BA and DHFR enzyme (**A**) from four types of fungi (**B**—*C. tropicalis*, **C**—glabrata, **D**—*A. flavus*, and **E**—*T. rubrum*). Hydrogen bonds are highlighted in dark green; van der Waals interactions in light green; hydrophobic interactions in pink and lilac; and steric interactions in red.

**Table 1 pathogens-12-00449-t001:** Information about the proteins selected in the study and the standard drugs used as controls.

Species	Protein	PDB ID/Homology	Medication	Resolution
	DNA gyrase	6FQM	ciprofloxacin	3.06 Å
*S. aureus*	beta-lactamase	1GHP	sulbactam	1.76 Å
	PBP	3VSL	imipenem	2.40 Å
	DNA gyrase	Homologia	ciprofloxacin	-
*S. epidermidis*	beta-lactamase	Homologia	sulbactam	-
	PBP	Homologia	imipenem	-
	DNA gyrase	6M1J	ciprofloxacin	1.70 Å
*P. aeruginosa*	beta-lactamase	5EPH	sulbactam	1.79 Å
	PBP	3PBQ	imipenem	1.70 Å
	DNA gyrase	5L3J	ciprofloxacin	2.83 Å
*E. coli*	beta-lactamase	1IEM	sulbactam	2.30 Å
	PBP	6G9P	imipenem	2.10 Å
	DNA gyrase	3IFZ	ciprofloxacin	2.70 Å
*M. tuberculosis*	beta-lactamase	3N7W	sulbactam	1.70 Å
	PBP	6KGW	imipenem	2.41 Å
	CYP	5FSA	fluconazole	2.86 Å
*C. albicans*	SAP-2	3PVK	pepstaine	1.27 Å
	DHFR	3QLW	trimethoprim	2.50 Å
	CYP	6T1U	fluconazol	1.50 Å
*C. tropicalis*	SAP-2	Homologia	pepstaine	-
	DHFR	Homologia	trimethoprim	-
*C. glabrata*	CYP	5JLC	fluconazole	2.40 Å
*A. flavus*	CYP	Homologia	fluconazole	-
	DHFR	6DRS	trimethoprim	2.00 Å
*T. rubrum*	DHFR	Homologia	trimethoprim	-

**Table 2 pathogens-12-00449-t002:** Minimum inhibitory concentration (MIC) and MBC of betulinic acid and the control (imipenem) against bacterial strains in µM.

Bacterial Strains	Betulinic Acid		Imipenem		Effect
Gram Positive	MIC	MBC	MIC	MBC	
*Staphylococcus aureus* ATCC-13150	561	1122	855	1710	Inhibitory
*S. epidermidis* ATCC-12228	561	1122	855	1710	Inhibitory
Gram negative	MIC	MBC	MIC	MBC	
*Pseudomonas aeruginosa* ATCC-25853	1122	2245	1710	3420	-
*Escherichia coli* ATCC-18739	561	2245	1710	3420	Inhibitory

The tests were performed in triplicate.

**Table 3 pathogens-12-00449-t003:** Minimum inhibitory concentration (MIC) and MFC of betulinic acid and the control (fluconazole) against fungal strains in µM.

Fungal Strains	Betulinic Acid		Fluconazole		Effect
Yeasts	MIC	MFC	MIC	MFC	
*Candida albicans* ATCC-90028	561	1122	835	1671	Inhibitory
*C. albicans* LM-34	561	1122	835	1671	Inhibitory
*C. tropicalis ATCC-13803*	561	1122	835	1671	Inhibitory
*C. glabrata* ATCC-90030	561	1122	835	1671	Inhibitory
Filamentous	MIC	MFC	MIC	MFC	Effect
*Aspergillus flavus* ATCC-13013	R	R	1671	3443	Resistant
*Penicillium citrinum* ATCC-40011	1122	2245	1671	3443	Inhibitory
*Trichophyton rubrum* LM-34	561	1122	835	1671	Inhibitory
*Microsporum canis* LM-12	561	1122	835	1671	Inhibitory

The tests were performed in triplicate.

**Table 4 pathogens-12-00449-t004:** Minimum inhibitory concentration (MIC) of betulinic acid and controls against *M. tuberculosis*.

Compounds	MIC in µM
betulinic acid	100
moxifloxacin	0.19
rifampin	0.15

The tests were performed in duplicate.

**Table 5 pathogens-12-00449-t005:** Binding energy values of BA against enzymes of selected bacterial species. Missing values are due to missing data and/or structures for performing molecular docking.

Species	DNA Gyrase		Beta-Lactamase		PBP	
	Betulinic Acid	Ciprofloxacin	Betulinic Acid	Sulbactam	Betulinic Acid	Imipenem
*S. aureus*	**−109.22**	**−96.78**	**−100.68**	**−69.29**	−82	−103.63
*S. epidermidis*	−73.89	−101.58	**−118.61**	**−68.33**	−76.51	−100.61
*P. aeruginosa*	−107.14	−121.73	**−112.71**	**−72.25**	**−107.59**	**−113.99**
*E. coli*	**−99.59**	**−89.57**	**−101.68**	**−68.35**	−71.63	−91.57
*M. tuberculosis*	**−114.43**	**−78.99**	**−86.77**	**−73.68**	−73.29	−100

The best results are highlighted in bold.

**Table 6 pathogens-12-00449-t006:** Binding energy values of BA against enzymes of selected fungal species. The best results are highlighted in bold. Missing values are due to missing data and/or structures for performing molecular docking.

Species	CYP51		SAP−2		DHFR	
	Betulinic Acid	Fluconazole	Betulinic Acid	Pepstaine	Betulinic Acid	Trimethoprim
*C. albicans*	**−120.98**	**−112.67**	−46.73	−84.10	−68.47	−82.96
*C. tropicalis*	**−113.31**	**−119.54**	−99.36	−129.96	**−87.25**	**−92.43**
*C. glabrata*	**−130.95**	**−77.29**	−	−	**−96.25**	**−83.97**
*Aspergillus flavus*	**−120.26**	**−125.17**	−	−	**−103.16**	**−87.71**
*Penicillium citrinum*	−	−	−	−	−	−
*Trichophyton rubrum*	−	−	−	−	**−151.48**	**−108.78**
*Microsporum canis*	−	−	−	−	−	−

The best results are highlighted in bold.

**Table 7 pathogens-12-00449-t007:** Description of the bonds and interactions observed between BA and the enzymes of the investigated bacteria and fungi.

Species	DNA Gyrase	
	Hydrogen Bonds	Hydrophobic Interactions
*S. aureus*	Glu88.	Arg458 and Glu477.
*E. coli*	−	Val43, His55, Arg76, Pro79, Val120, Val167.
*M. tuberculosis*	−	Pro42, Leu48, Lys49, His52, Arg98, Leu105 e Tyr276.
**Species**	**Beta−Lactamase**	
*S. aureus*	Gln237.	Ala69, Tyr105, Asn132, and Ile239.
*S. epidermidis*	Gln435 and Thr529.	Met434, Ile515, Lys526, Phe541 e Lys566.
*P. aeruginosa*	Thr159 and Thr228.	Cys61, Tyr97 e Pro160.
*E. coli*	Ser64 and Ala318.	Tyr221 and Leu293.
*M. tuberculosis*	Ser84, Thr251, and Asp255.	Ile117 e Arg187.
**Species**	**PBP**	
*P. aeruginosa*	Asn351.	Val333, Lys348 and Tyr532.

**Table 8 pathogens-12-00449-t008:** Description of the bonds and interactions observed between BA and the enzymes of the investigated bacteria and fungi.

Species	CYP51	
	Hydrogen Bonds	Hydrophobic Interactions
*C. albicans*		Tyr118, Phe228, Pro230, Leu376, Hist377, Phe380 and Met508.
*C. tropicalis*	Tyr 156.	Thr67, and Tyr118.
*C. glabrata*	−	Tyr127, Leu130, Phe135, Phe237, Met314, Phe385, Leu381, His382, and Met512.
	His207 and Ser495.	Ala178, Ile182, Tyr206, His295, Leu290, Met291, and Pro497.
**Species**	**DHFR**	
*C. tropicalis*	Ala12 and Ile113.	Ile10, Met26, Ile34, Tyr36, Ile63, and Pro64.
*C. glabrata*	Ile121 and Tyr127.	Val10, Ile19, Leu25, Trp27, Met33, Ile62, Phe66, and Leu69.
*A. flavus*	−	Val11, Ala12, Ile26, Leu32, Met41, Phe44, Val70, Leu74 e Leu77.
*T. rubrum*	−	Ala13, Ile30, Leu36, Phe48, and Met182.

## Data Availability

Not applicable.

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
