# Peer review of "Antimicrobial Potential of Betulinic Acid and Investigation of the Mechanism of Action against Nuclear and Metabolic Enzymes with Molecular Modeling"

_pathogens, 2023, doi:10.3390/pathogens12030449_

Round 1

Reviewer 1 Report (Previous Reviewer 3)

After reading the corrected manuscript it seems the authors have made the suggested changes from the 3 reviewers.

Author Response

Reviewer 2 Report (Previous Reviewer 1)

Paper titled (In vitro evolution of the antimicrobial potential of betulinic acid and analysis of mechanisms of action with molecular modeling) by Rodrigues etal. tested the antimicrobial potential of betulinic acid in vitro and perfromed a molecular modeling study for the putative mechanism of action.

1- Title: can be amneded by mention the mechanism of action and to be more informative

2- Abstract can be amended by some numerical values

3- Introduction: combine  paragraph1 & 2 - Paragraph 3 better to come at the begin of INTRO

4- Aim: mention the rational, why authors thought that BA will have antimicrobial activity

5-  Method of extraction should be given in very detailed procedures. Did the authors used an alaytical or HPLC standard to validate the method?

6- Table 2: how many times the MIC was calculated for each strain ? mention in the table footnote so the authors can evalutae the strenghth of the study results

Authors should measure the MIC for at least  3 times, one time (& calculate mean & SD) is not meaningful

7- The same for Table 3 & Table 4

8- Conclusions are too long & needs to be more concrete and discuss the findings of the current study only

Author Response

Reviewer 3 Report (New Reviewer)

Please find attached

Round 2

Reviewer 2 Report (Previous Reviewer 1)

Unfoutnately, the revised version of the manuscript was not revised according to the reviewer's comments. Method of extraction is not clear!! what was the extracted compound(s) !!! only one tested sample is not enough to draw a conclusion. I completly recommend against publication of this manuscript.

Author Response

Reviewer 3 Report (New Reviewer)

Thank you for the additional details and answers. The requested completions have been made. The article can be accepted for publication, but please note the following:

- line 19: the term botulinic acid should be corrected to betulinic acid

- line 410, Figure 4. 3D interactions between compound BA and the beta-lactamase enzyme (A) of three bacterial species (B – S. aureus, S. epidermidis, P. aeruginosa The above description is incomplete, it is not indicated what figures C, D and D, E represent.  

Author Response

This manuscript is a resubmission of an earlier submission. The following is a list of the peer review reports and author responses from that submission.

Round 1

Reviewer 1 Report

Paper titled (in vitro evolution of the antimicrobial potential of betulinic acid and analysis of mechanisms of action with molecular modeling) by Soares Rodrigues et al. is an article screened the antimicrobial potential of betulinic acid & tried to explore the mechanism of action by molecular modeling. This is a novel study with a straightforward aim. However, needs to be extensively revised. I have the following recommendations for improvement.

1- Title: I find the word " evolution" is not necessary or otherwise the authors may convince me about its value.

Try to avoid "analysis" of docking & choose another word

2- Abstract : can be amended by some  numerical values

3- Figures should be reorganized & panels should be labeled (A,B,C,........etc) & describe each panel in the figure legend.

4- Introduction: in the last part "aim" describe which fungal or bacterial strains were used; better to specify which strains. or at least a common feature between them.

5- Methods: he method of extraction of Betulinic acid was extracted from the bark of Zizyphus joazeiro Mart. (Rhamnaceae) 84 using a method described earlier" should be written in details and how you identified BA?

Otherwise, we cannot rely on results of this study.

& then, mention in title & everywhere, authors used an extract of ........... (instead of BA).

6- This reviewer noticed that all data in Tables are single experiiment as there is no SD for the results. Mention this in methods & Table foot notes, so the reader can evaluate the integrity of these results.

7- Methods in general lacks references at many occasions

8- Authors should give the source of chemicals, kits and antibodies completely and consistently (code, company, town, state and country) & version for software

Reviewer 2 Report

The study by Rodrigues et al. aimed to investigate the activity of betulinic acid against M. tuberculosis, E. coli, and fungi in invitro using MIC and MBC. Further, molecular modeling studies were carried out to investigate the compound's mechanism of action. The results showed that the compound has antimicrobial activity, at a concentration of 256 μg/mL, inhibited the growth of 75% of the microbial species, and the mechanisms of action are due to DNA gyrase and beta-lactamase targets in bacteria and targeting sterol 14α-demethylase (CYP51), and dihydrofolate reductase (DHFR) in fungi.

This study is relevant and important due to the rising antimicrobial resistance. A new alternative antimicrobial is needed. However, there are major  problems with this manuscript

1.    The documents cited for MIC ( the protocols of Cleeland and Squires 1991  and Eloff 1998 ) are largely outdated. The MIC should be performed based on standardized methods such as CLSI, and no control positive was used. As a consequence, the MIC data is not valid

2.    The reference format and citation  used are largely out of journal style

3.    Line 86: section 2.2: the subheading sentence should be written in the English language

4.    The bacteria name should be written in italic throughout the manuscript

Reviewer 3 Report

Comments about grammatical issues:

- The whole text looks like it has been written in another language (Portuguese, I guess) and translated by one of those automatic robot translators. The whole text from line 1 to the last one needs to be read and corrected by an English native speaker. In fact, some parts are still in original Portuguese: eg. Line 86.

- In the same fashion, tables 2 and 3 must be corrected. They show MIC and MBC in captions and CIM, and CBM on the tables, for instance. Same with CFM and MFC.

- Authors call betunilic acid throughout the whole text sometimes as betulinic acid, compound BA, the BA molecule, Triterpene BA or simply BA. Suggestion: chose one way to call the compound and use that way on the whole text unless is important to use a different name.

- Lines 23-24 and 30-31 have the same information

- Line 24: 75% fungi; line 31 75% fungus

- Line 38 word “roots” is repeated twice

- Line 42:’ BA has gained much concentration…” Does it mean attention?

- Figure 1 has plots called small case (a) and (b) and captions are in big case (A), (B)

- Reference 3 is all capitalized

- And so on. It is not necessary top show more of these typos and grammatical issues. They are everywhere.

Comments about the science of the paper:

- The abstract starts with the statement that: “Natural products are considered important therapeutic agents, so the present study sought to investigate the activity of betulinic acid against different strains of bacteria and fungi”. Comment: natural products are NOT intrinsically therapeutic agents. They can be many other things so just for being “natural” they are not automatically therapeutic. This need to be rephrased.

- Lines 76- 77 the authors state:” This information significantly contributes to reducing costs in the production of antimicrobial drugs that have low potential”. If a drug has low potential, why does anyone will produce it no matter the costs? I think we can see what the authors try to say but needs clarification.

- Though the molecular structure of betunilic acid is shown several times in Figures 3, 4 and others, these figures are modified by captions and additions and they are confusing. Suggestion: create a figure with a clean and clear view of the structure of betulinc acid. In those showed figures there are some parts missing like hydrogens on two OH groups, I think, since we do not have that clean structure. There is no stereochemistry shown at all. What are the spatial position of the methyl groups? Are they methyl groups? There are some histidine interactions with ring A. Is this ring aromatic? I think not (too many bonds) but those, and other, interactions look odd without any structural context.

- The antimicrobial results are done in vitro with actual bacteria and the results show some inhibitory activity (tables 2 and 3), however, the authors do molecular docking of BA with some potential protein targets and use that to explain such activity because these targets seem to be common for the bacteria tested. This is a big assumption that the targets of betulinic acid on the tested bacteria are actually those calculated/docking enzymes. It is fine as a good and educated guess but it is not conclusive that that is what it happens. If the authors can make a real statement perhaps they should do an actual kinetic experiment with those enzymes and BA to test if they actually get inhibited. Otherwise their conclusions must be approximate.

- About conclusions, they are as long as the discussion section. I would suggest that some of the comments in conclusions to be moved up to discussion.

The work itself is interesting and as mentioned, the experimental section can be useful for someone working in this field but as noticed above, the whole text needs a better translation, text corrections and rewording, clarifications or explanation of some statements done.